# Astaxanthin Supplementation Reduces Subjective Markers of Muscle Soreness following Eccentric Exercise in Resistance-Trained Men

Gaven A. Barker [1], Alyssa L. Parten [1], David A. Lara [1], Kensey E. Hannon [1], Matthew J. McAllister [2] and Hunter S. Waldman [1,*]

[1] Department of Kinesiology, University of North Alabama, Florence, AL 35632, USA
[2] Metabolic & Applied Physiology Lab, Department of Health and Human Performance, Texas State University, San Marcos, TX 78666, USA
* Correspondence: hswaldman@una.edu

**Abstract:** Strenuous exercise involving eccentric muscle actions induces skeletal muscle damage resulting in delayed onset muscle soreness (DOMS). Antioxidant supplementation, such as astaxanthin (AX), may alleviate muscle injury following intense exercise. The purpose of this study was to investigate the effect of a four-week course of AX supplementation at $12 \text{ mg/day}^{-1}$ on subjective markers of DOMS, recovery, and performance after a bout of muscle damaging eccentric exercise. Nineteen resistance-trained men (mean ± SD: age, 22.6 ± 2.2 y) completed a between-group design with a four-week supplementation period of $12 \text{ mg/day}^{-1}$ of either AX or a placebo. Subjects completed four trials, with trials One and Three designed to induce muscle damage, consisting of a one repetition maximum test (1RM) for leg-press, followed by five sets of ten repetitions at 65% of 1RM. Trials Two and Four were performance trials, conducted 48 h later and consisting of repetitions to failure at 65%, 70%, and 75% of 1RM. Subjective markers of DOMS and recovery were collected at multiple timepoints post-trial for trials One and Three. Although performance was not affected ($p > 0.05$), AX supplementation significantly decreased subjective markers of DOMS ($p = 0.01$) compared to the placebo. The results demonstrated that AX may enhance recovery by reducing DOMS without detriment to performance in resistance-trained men.

**Keywords:** antioxidant; muscle damage; recovery; supplement; performance



## 1. Introduction

It is well established that strenuous exercise, predominantly involving eccentric muscle contractions, induces skeletal muscle damage resulting in delayed onset muscle soreness (DOMS) [1–3]. The symptoms of DOMS can vary from light muscle stiffness to severe pain and muscle soreness which may restrict movement, alter joint kinematics, reduce force production (~21 days), and alter muscle fiber recruitment patterns, and can increase an individual's risk of injury [4]. DOMS typically occurs in as little as 8–10 h post exercise [4] and reaches its peak around 48–72 h post eccentric-focused exercise [5]. Eccentric activity, in particular, is hypothesized to result in significantly greater muscle damage, as it elongates the muscle during simultaneous contraction which increases myofibrillar stress and Z-disk disruption. There are several proposed mechanisms that attempt to explain the pain stimulation of DOMS including the aforementioned increase in muscle damage, increase in inflammation, enzyme influx theories and the excessive production of reactive oxygen species (ROS) [4]. While no singular theory can be attributed as the sole cause of DOMS, it appears that the excessive production of ROS is one of the most consistently suggested explanations [6]. In brief, excessive ROS production resulting in oxidative stress creates an imbalance between the production of ROS and the rate of its removal, particularly through the counteractivity of the cellular antioxidative systems [7,8]. While some ROS is

beneficial for the promotion of cellular adaptations from exercise [9], excessive ROS from intense and eccentric exercise may elevate DOMS and, thus, impair performance [10,11]. Therefore, interventions which may mitigate DOMS or excessive ROS production, such as supplementing with dietary antioxidants, are of great interest to coaches and athletes alike.

Dietary supplements are often used following intense exercise in an attempt to accelerate the recovery process [12,13]. Dietary antioxidants are one supplement that may accelerate recovery by mitigating DOMS [7,14], likely through the reduction of oxidative stress [15]. Despite indication of the potential mechanisms for antioxidant supplementation to reduce DOMS, the evidence has been mixed, possibly due to differences in methodology [16–20]. The carotenoid *Haematococcus pluvialis*, commonly known as astaxanthin (AX), has been suggested to exhibit antioxidative properties superior to most other antioxidant agents, such as vitamins C or E [21]. AX is a lipid soluble carotenoid found primarily in microalgae and marine species [22]. The unique chemical structure of AX allows it to permeate through the phospholipid bilayers of the mitochondria, acting as a direct scavenger of inter- and extra-cellular ROS production [23]. Interestingly, recent evidence has shown that AX increases glutathione concentrations [24], one of the cell's most robust endogenous antioxidants. However, only a few studies have attempted to examine the impact of AX on markers of exercise-induced muscle damage (EIMD) and DOMS [1,25,26]. The first of these studies reported no changes in muscle soreness or creatine kinase (CK), a blood-borne marker for skeletal muscle damage, after eccentric-focused exercise following AX supplementation of 8 mg/day$^{-1}$ for 3-weeks [1]. Similarly, Waldman et al. [26] recently reported that a four-week course of AX supplementation (12 mg/day$^{-1}$) did not impact markers of muscle damage, inflammation, or DOMS after an EIMD protocol. Conversely, Baralic et al. [25] reported AX supplementation (4 mg/day$^{-1}$) for a 90-day period attenuated markers of muscle damage. The discrepancies in these findings may be affected by differences in several variables, such as the muscle damaging protocols adopted, markers analyzed for DOMS and inflammation, and the dosage of AX ingested.

The aforementioned studies [1,25,26] examined various blood markers to measure DOMS, sometimes with subjective measurements as a secondary purpose of the investigation. While measuring inflammation or DOMS via markers of damage found in blood or muscle tissue are common, these techniques can further exacerbate the inflammatory process by the insertion of a needle [27]. Interestingly, noninvasive measures, such as visual analog scales (VASs), have been used in numerous studies as a subjective measurement of DOMS [26,28–33] and, in research examining antioxidant supplements similar to AX, VAS was shown to increase linearly when several objective measures of muscle damage also increased [28–30]. The convenience of assessing subjective measures makes the use of such measures more practical in a variety of sport or exercise settings for the coach or athlete, and, furthermore, the collection of noninvasive measurements, such as perceived recovery scores (PRS), and rating of perceived exertion have implications for exercise performance [12,34]. Past research has reported that an increase in DOMS reduces exercise performance [4], and is mainly associated with a reduction of force output [35]. Thus, the primary purpose of this study was to investigate the effect of AX supplementation at 12 mg/day$^{-1}$ for four weeks on various subjective markers of DOMS following a bout of extensive muscle damaging eccentric exercise. A secondary purpose was to examine if AX impacted markers of resistance training performance.

## 2. Results

Descriptive statistics by group and variable are presented in Table 1. Statistical analyses revealed no significant differences for group × condition × time ($p = 0.81$), group × time ($p = 0.44$), or condition × time ($p = 0.13$) for SORE. However, significance was found for group × condition ($p = 0.01$; $\eta_p^2 = 0.65$), with the AX group demonstrating a significant reduction in SORE from pre- to post-supplementation ($p = 0.01$) assessments. Similarly, VAS displayed no significant differences for group × condition × time ($p = 0.41$) group × time ($p = 0.32$), or condition × time ($p = 0.62$), but did have a significant difference

in group $\times$ condition ($p = 0.02$; $\eta_p^2 = 0.73$; Table 2) with the AX group experiencing a significant decrease in VAS from pre- to post-supplementation ($p = 0.009$) assessments. Moreover, there were no significant main effects or interactions in markers of performance at any intensity (65–75%; all $p > 0.05$; Table 3). For PRS, a significant time effect was observed ($p < 0.001$; $\eta_p^2 = 0.25$), though no significant group effects or interactions were observed ($p > 0.05$). Overall, both groups experienced a significant decrease in PRS from the EIMD session to the performance trial, which was noted for the pre-and post-supplementation conditions ($p < 0.001$). Further, no significant main effects or interactions were found in SRPE at any respective timepoint (all $p > 0.05$; Table 4).

**Table 1.** Anthropometric characteristics between groups ($n = 19$; mean $\pm$ SD).

|  | **Placebo ($n = 9$)** | **Astaxanthin ($n = 10$)** |
|---|---|---|
| Age | $23.1.0 \pm 2.0$ | $22.0 \pm 2.4$ |
| Body mass (kg) | $93.9 \pm 13.8$ | $91.6 \pm 9.8$ |
| Height (cm) | $180.2 \pm 4.0$ | $178.6 \pm 6.0$ |
| Body fat (%) | $22.4 \pm 5.8$ | $22.6 \pm 6.6$ |

**Table 2.** Various subjective reports of DOMS ($n = 19$; mean $\pm$ SD).

|  | **24-h Pre** | **24-h Post** | **36-h Pre** | **36-h Post** | **48-h Pre** | **48-h Post** |
|---|---|---|---|---|---|---|
| SORE (0–6) |  |  |  |  |  |  |
| AX ($n = 10$) * | $3.3 \pm 0.8$ | $1.4 \pm 1.1$ |  |  | $2.8 \pm 1.2$ | $1.9 \pm 1.2$ |
| PLA ($n = 9$) | $2.8 \pm 1.5$ | $3.0 \pm 1.4$ |  |  | $2.3 \pm 1.2$ | $2.5 \pm 1.6$ |
| VAS (mm) |  |  |  |  |  |  |
| AX ($n = 10$) ** | $69.9 \pm 62.3$ | $28.2 \pm 18.8$ | $68.0 \pm 27.5$ | $28.2 \pm 19.6$ | $63.1 \pm 31.2$ | $24.9 \pm 19.8$ |
| PLA ($n = 9$) | $62.3 \pm 32.1$ | $70.9 \pm 40.1$ | $56.9 \pm 36.7$ | $63.7 \pm 34.1$ | $55.9 \pm 31.4$ | $54.8 \pm 30.0$ |

* Denotes a significant reduction in post- vs. pre-soreness ($p = 0.01$). ** Denotes a significant reduction in post- vs. pre- VAS ($p = 0.02$). Pre = after exercise-induced muscle damage (EIMD) protocol before supplementation; Post = after 4 weeks of supplementation and post-EIMD protocol; AX = astaxanthin; PLA = placebo; DOMS = delayed onset muscle soreness; VAS = visual analog scale.

**Table 3.** One repetition maximum and performance trial repetitions ($n = 19$; mean $\pm$ SD).

|  | **Pre** | **Post** | **Pre** | **Post** |
|---|---|---|---|---|
|  | AX ($n = 10$) | AX ($n = 10$) | PLA ($n = 9$) | PLA ($n = 9$) |
| 1RM (kg) | $352.9 \pm 75.2$ | $379.3 \pm 75.2$ | $387.1 \pm 66.5$ | $401.5 \pm 67.3$ |
| 65% 1RM (reps) | $19 \pm 6$ | $20 \pm 8$ | $21 \pm 7$ | $23 \pm 12$ |
| 70% 1RM (reps) | $13 \pm 5$ | $12 \pm 4$ | $16 \pm 5$ | $16 \pm 7$ |
| 75% 1RM (reps) | $10 \pm 3$ | $9 \pm 4$ | $11 \pm 3$ | $14 \pm 7$ |

**Table 4.** Session rating of perceived exertion and perceived recovery scale ($n = 19$; mean $\pm$ SD).

|  | **SRPE** |  | **PRS** |  |
|---|---|---|---|---|
|  | **Pre** | **Post** | **Pre** | **Post** |
| EIMD |  |  |  |  |
| AX ($n = 10$) | $8.6 \pm 0.8$ | $8.3 \pm 0.7$ | $8.6 \pm 0.8$ | $8.4 \pm 1.5$ |
| PLA ($n = 9$) | $7.4 \pm 1.0$ | $7.8 \pm 0.7$ | $9.0 \pm 1.1$ | $8.5 \pm 1.9$ |
| PT |  |  |  |  |
| AX ($n = 10$) | $7.3 \pm 0.9$ | $7.8 \pm 1.1$ | $6.8 \pm 2.0$ | $7.7 \pm 0.8$ |
| PLA ($n = 9$) | $7.3 \pm 0.9$ | $7.0 \pm 1.1$ | $6.6 \pm 1.8$ | $6.7 \pm 1.9$ |

EIMD = exercise-induced muscle damage trial; PT = performance trial; AX = astaxanthin; PLA = placebo; SRPE = session rating of perceived exertion; PRS = perceived recovery score.

## 3. Discussion

The primary purpose of this study was to examine the effects of four weeks of AX supplementation on subjective markers of DOMS and recovery, as well as resistance exercise performance. Our data showed that, compared to PLA, subjects who supplemented with

12 mg/day$^{-1}$ of AX experienced significant decreases in subjective markers of DOMS and soreness after an EIMD protocol. These findings contradict the findings of Bloomer et al. [1] and Waldman et al. [26], who demonstrated that three weeks of AX supplementation at 4 mg/day$^{-1}$ and four weeks of AX supplementation at 12 mg/day$^{-1}$ did not mitigate markers of muscle soreness or damage after eccentric exercise protocols, respectively. Additionally, our secondary findings showed that four weeks of AX supplementation did not impact exercise performance in repetitions to failure during leg press at 65%, 70%, and 75% of 1RM. These results collectively demonstrate that the properties of AX can reduce DOMS without inhibiting resistance training performance in resistance-trained men.

Extensive research has been conducted examining the role antioxidants have in the enhancement of exercise recovery by mitigating DOMS, although data are mixed [7,14,15,36]. The primary findings of the present study showed that AX supplementation significantly reduced subjective muscle soreness after an EMID protocol in resistance-trained males. There was a significant decrease in both SORE ($p$ = 0.02) and VAS ($p$ = 0.01) for the AX group, while, concurrently, no observed differences in the PLA group were evident (see Table 2). Our primary objective was to assess noninvasive and subjective measures of DOMS following supplementation, and, on this basis, the present study provides practical findings for the coach or athlete interested in mitigating DOMS. During intense competition or training blocks, teams and individuals alike may find interventions which reduce subjective sensations of soreness favorable, especially as alternative measures, such as blood biomarkers or dynamometers, may not be practical in these settings. However, we also acknowledge that our study lacked these objective markers of muscle damage, which limited understanding of the exact mechanisms by which AX acted to mere speculation. While the study's findings suggest that AX supplementation most likely lowered inflammation that was, in part, attributed to an increase in ROS production [22], AX may have also mitigated myotube damage, compared to PLA alone, as AX supplementation has shown a plethora of beneficial effects regarding muscle health, including enhanced myocyte membrane stability [37]. Nonetheless, the mitigation of the subjective sensation of DOMS has practical applications in reducing the perception of physical impairment [4] and the sensation of pain [14].

Discrepancies between past research [26] and the present study may be attributed to differences in the protocol for inducing EIMD. While the EIMD protocol used by Waldman et al. [26] achieved *statistical* significance with respect to DOMS, the actual magnitude of change was small ($\eta_p^2$ = 0.42). Therefore, we sought to adopt a protocol shown to induce a high magnitude of change from baseline to markers of DOMS in a similarly trained cohort [13]. Given our contrasting data to the findings of Waldman et al. [26], and the moderate–large effect we observed following the EIMD protocol ($\eta_p^2$ = 0.65), future studies may consider incorporating a similar EIMD protocol as the present study when examining the impact of antioxidant supplementation and markers of DOMS or soreness.

With respect to exercise, most research on AX has examined either endurance performance [22,38–40] or changes in markers of substrate oxidation rates [24]. Collectively, AX has been shown to reduce average heart rate at submaximal endurance intensities [39], improve cycling time trial performance [38,40], and in one study, enhance whole body fat oxidation rates [38]. Despite these findings, fewer studies have attempted to assess the role of AX in anaerobic performance. It has been previously hypothesized that AX may negatively impact resistance training performance by limiting training adaptations due to the antioxidant properties it exudes [41]. However, the present study observed no significant differences between groups following supplementation for repetitions to failure at 65%, 70%, or 75% of the subjects' 1RMs. These results suggest that AX decreased subjective markers of DOMS without impairment to resistance exercise performance. Moreover, the present study also observed no changes to PRS or SRPE. While it is only speculation, the lack of change to PRS may be explained by simple differences in timepoints at which PRS and DOMS were collected. *Post*-hoc analysis did not reveal specifically when DOMS was lowered, rather only that the AX group experienced significantly less DOMS post-

supplementation overall compared to the PLA group. The significant change in DOMS may have occurred at an earlier timepoint than that when PRS was assessed. Further, it is possible that the collection of PRS at additional timepoints may have revealed a different finding. This is a limitation to the present study and should be addressed in future research. Similarly, the lack of SRPE group differences appears to reflect the timepoint at which data were collected. Given that SRPE was collected 15 min post-trial, it is unlikely that any meaningful sensation of DOMS had occurred between either group at this timepoint.

The present study is not without limitations. While the aim of this study was to investigate the impact of AX supplementation on subjective markers of DOMS and recovery, it should be noted that no objective measures of DOMS were collected, and, therefore, the discussion of our observed changes is limited to speculation. This, however, provides an area for future studies as to whether changes in subjective markers from AX supplementation correlates with changes in blood biomarkers or more sophisticated measures of DOMS, such as reductions in isometric strength. Additionally, although our subjects were asked to continue their habitual training and diets, the present study did not control for these variables during the four-week supplemental periods. Controlling for total training load over the entire supplementation period would have better isolated the effects of AX supplementation on performance and should be addressed in future studies.

In conclusion, our findings suggest that four weeks of AX supplementation ($12 \, \text{mg/day}^{-1}$) significantly reduced subjective perceptions of DOMS following eccentric exercise, when compared to a PLA group. Furthermore, AX did not impact resistance training performance, suggesting that AX supplementation may be an intervention resistance-trained males may consider for alleviating perceived soreness during periods of intensive training. Future research should consider examining AX supplementation in a female-trained cohort, as the research is currently lacking with respect to females and AX supplementation on markers of performance or metabolism.

## 4. Materials and Methods

### 4.1. Subjects

Twenty-two resistance-trained men were initially recruited to participate as subjects in the present study. However, three subjects were dropped due to injury from their habitual training cycles and, thus, nineteen subjects completed the study in its entirety ($n$ = 19; Table 1). To be considered resistance-trained, subjects were required to be currently attending ≥3 sessions of resistance training per week. Furthermore, subjects had to have been resistance training continuously for ≥4 months and to have accumulated at least 2 h per week of resistance-based exercise to be included. Other inclusion criteria required each subject to meet the American College of Sports Medicine low-risk guidelines [42], to not currently be competing in an organized collegiate or professional sport, and to refrain from all dietary supplement use (e.g., multi-vitamins, creatine, and antioxidants) for at least two weeks prior to testing. This population was chosen based on its similarity to the populations of other studies that investigated AX effects on muscle soreness and because of their potential to most likely complete the volume and intensity necessary to induce DOMS [1,26] A list of AX-rich foods (e.g., salmon, crustaceans, etc.) was provided to the subjects prior to testing. Subjects were asked to avoid these items during both 4-week supplementation periods. Subjects were instructed to refrain from caffeine and alcohol 24 h prior to testing, to arrive well-hydrated (not assessed), and also to complete a 24-h dietary and physical activity log to ensure diet and exercise were the same for pre- and post-testing. It is important to note that the investigative team did not control the subjects' training during the 4-week supplemental periods. However, each subject was asked to continue their habitual exercise regimen across the 4 weeks and to avoid any strenuous exercise 48 h prior to each experimental session. These parameters were verbally confirmed by all subjects prior to each trial. In addition, all subjects were recruited via word of mouth and were given explanations of all procedures, risks, and benefits, and, subsequently, asked to provide their written and verbal consents to participate in the present study, which

was approved by the University's Institutional Review Board (IRB#: 2022-009) and was in accordance to the Declaration of Helsinki.

Procedures

The present study followed a placebo (PLA)-controlled, double-blind (subjects and co-investigator administering the AX), 2-arm parallel design to examine the effects of an AX supplement on subjective markers of DOMS and recovery, and resistance training performance. Subjects reported to the laboratory on 5 separate occasions (see Figure 1): one initial session to collect preliminary data and washout any potential learning effects (familiarization), two trials pre-supplementation (trials 1 and 2), and two trials post-supplementation with either AX or PLA (trials 3–4).

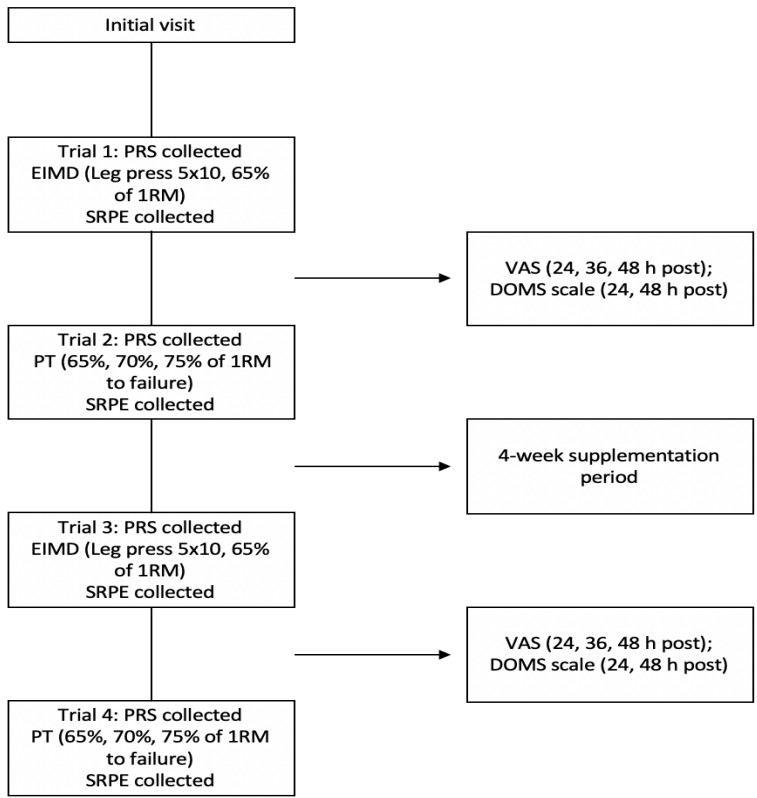

**Figure 1.** Overview of study methodology. EIMD = exercise-induced muscle damage; 1RM = 1 repetition maximum; PRS = perceived recovery score; SRPE = session-rating of perceived exertion; VAS = visual analog scale; DOMS = delayed-onset muscle soreness; PT = performance trial.

Upon arrival at the laboratory for their initial visits, each subject completed written informed consent, a physical activity readiness questionnaire, and an exercise history questionnaire to ensure the inclusion criteria were met. Additionally, each subject had his body mass (Tanita Corporation, Tokyo, Japan), height (Deteco, Webb City, MO, USA), and body fat (%) assessed using bioelectrical impedance analysis (SECA mBCA 514, SECA North America, Chino, CA, USA). All testing was conducted at the same time of day for each subject and on the same leg press (Hammer Strength Plate-Loaded Linear Leg Press, Life Fitness, Chicago, IL, USA). Prior to each trial, subjects were asked how recovered they felt, based on a visual PRS (0–10) [43], with 0 reflecting "very poorly recovered" and 10 reflecting "very well recovered". All subjects then completed the same warm-up consisting of a 400 m jog, one set of 10 repetitions on an empty leg press (53 kg), and one set of 10 repetitions with additional resistance (94.5 kg). Moreover, following each trial, a session rating of perceived exertion (SRPE) was assessed 15 min after the completion of the last repetition, using an OMNI pictorial scale to measure each subject's perceived effort given throughout the entirety of the session [35].

### 4.2. Exercise-Induced Muscle Damage (Trials 1 and 3)

After completion of the warm-up, each subject completed a muscle damaging protocol adopted from a previous study [13]. The protocol was modified from a bench press to a leg press to ensure safety, and also to induce greater sensation of muscle damage, based on our pilot testing prior to data collection. The trial consisted of finding a three-repetition maximum (3RM) of leg presses for each subject, which was then used to estimate their one-repetition maximum (1RM; load (kg)/(1.0278 − 0.0278 × 3 9; 36). Between each attempt, subjects were given a 5-min rest period. Following the 3RM testing, each subject was given a 15-min rest period prior to the start of the muscle damaging protocol, which consisted of five sets of 10 repetitions at 65% of their estimated 1RM, with a 3-s eccentric lowering phase using a metronome. Each subject completed a strict 3-min passive rest in-between sets. Following trials 1 and 3, DOMS was assessed in two different ways with direct supervision from a member of the research team. The first assessment was a VAS (150 mm) assessed post-EIMD at 24, 36, and 48 h, with 0 mm representing "no soreness" and 150 mm being "worst possible soreness", and then by a numeric soreness scale (SORE, 0–6; 37) at 24 and 48 h post-EIMD with 0 representing "no soreness" and 6 being "intolerable soreness". Subjects were then stratified into groups (i.e., AX or PLA), based on the trial 1, 48-h post-VAS score. In brief, subjects were numerically ranked highest to lowest according to their 48-h VAS score. The subject with the highest 48-h post VAS score was then assigned to one group, while the subject with the next highest 48-h post VAS score was assigned to the opposite group. This pattern continued until all subjects were assigned to either the AX or PLA group. The stratification was conducted by an independent investigator not involved in the data collection. An independent *t*-test revealed no significant group differences between baseline VAS scores ($p = 0.75$), which was a primary variable of interest in the present study.

### 4.3. Performance Trials (Trials 2 & 4)

Exercise performance trials occurred 48 h after trials 1 and 3. Following a standardized warm-up, each subject completed three sets of as many repetitions as possible until failure at 65%, 70%, and 75% of their estimated 1RM, with a passive 3-min rest between sets. In order for a repetition to be counted, the subject had to reach a depth of at least 90°, which was standardized during the familiarization session and monitored for consistency by the same two researchers for each performance trial.

### 4.4. Supplementation Schedule

Previous studies [1,19,24–26,38] have implemented varying doses of AX (4–40 mg/day$^{-1}$), with the most common dosages being 12 mg/day$^{-1}$ [25,34,44]. In addition, AX has been shown to reach peak plasma concentrations following 6 days of supplementation at ~1.25 mg/day$^{-1}$ [44]. Therefore, the present investigation incorporated 12 mg/day$^{-1}$ and a supplementation period of 4 weeks was adopted. Following trials 1 and 2, subjects were assigned to either the AX or PLA group, based on the stratification method detailed earlier. The AX and PLA capsules were provided by AstaReal (AstaReal, Inc., Moses Lake, WA, USA), and were matched in shape, size, and color, and were odorless. The same independent investigator, who stratified the groups, counted and distributed the pills into bottles labeled "A" or "B". Each subject consumed two capsules each day, one in the morning and one at night. Each capsule contained 6 mg of either AX or PLA (sunflower oil) for a total of 12 mg/day$^{-1}$. If an ingestion period was missed, subjects were instructed to take the missed capsule as soon as possible. Additionally, subjects were provided with an undisclosed, but specific, number of capsules in each bottle. Compliance ([capsules ingested]/56) × 100) was measured when the subjects returned the remaining capsules upon trial 4. Compliance less than 90% was defined as "not acceptable", requiring the subject to be removed. However, no subjects fell below this threshold and, thus, no subjects were removed from this study.

*4.5. Statistical Analyses*

Data are presented as the mean $\pm$SD. An a priori $\alpha$ level of $p \leq 0.05$ was used to determine significance in all analyses. A $2 \times 2 \times 3$ (group [AX vs. PLA] $\times$ condition [pre vs. post] $\times$ timepoint) mixed-factor, repeated-measures analysis of variance (ANOVA) was used to assess changes for VAS at each respective timepoint. A $2 \times 2 \times 2$ mixed-factor, repeated-measures ANOVA was used to assess changes to SORE, PRS, and SRPE, and a $2 \times 3$ mixed-factor, repeated-measures ANOVA was conducted to analyze performance at each respective intensity (65%, 70%, and 75%), pre- and post-supplementation. Mauchly's test was used to determine if acceptable ($p > 0.05$) sphericity was exhibited. If sphericity was infringed, the Greenhouse–Geisser technique was implemented to adjust degrees of freedom for main effects. Effect sizes were calculated and reported as partial eta-squared where significance occurred ($\eta_p^2$: <0.5 = small effect; 0.5–0.8 = moderate effect; and >0.8 = large effect). Bonferroni *post* hoc corrections were implemented when an interaction or main effect was observed. IBM SPSS Statistics v. 27 was used for all analyses.

**Author Contributions:** Conceptualization, G.A.B., M.J.M. and H.S.W.; Data curation, G.A.B., K.E.H. and H.S.W.; Formal analysis, G.A.B., A.L.P. and H.S.W.; Investigation, G.A.B., A.L.P., D.A.L., K.E.H. and H.S.W.; Methodology, G.A.B., D.A.L., M.J.M. and H.S.W.; Project administration, G.A.B. and H.S.W.; Supervision, H.S.W.; Writing—original draft, G.A.B., A.L.P., D.A.L., M.J.M. and H.S.W.; Writing—review & editing, G.A.B., A.L.P., D.A.L., K.E.H., M.J.M. and H.S.W. All authors have read and agreed to the published version of the manuscript.

**Funding:** This research received no external funding.

**Institutional Review Board Statement:** The study was conducted in accordance with the Declaration of Helsinki, and approved by the Institutional Review Board of the University of North Alabama (IRB#: 2022-009).

**Informed Consent Statement:** Informed consent was obtained from all subjects involved in the study and written consent was obtained for the data to be published from all subjects.

**Data Availability Statement:** The data presented in this study are available on request from the corresponding author.

**Acknowledgments:** The authors thank those who participated in this study, the University of North Alabama strength and conditioning staff, including Steve Herring, and all who helped with the collection of data. The authors also would like to thank AstaReal Inc., for their donation of the supplements and Karen Hecht for her input on the dosage protocol implemented. No authors derived financial benefit from the results of this research, and all authors declare no known conflict of interest.

**Conflicts of Interest:** The authors declare no conflict of interest.

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
