# Peer review of "Astaxanthin Supplementation Reduces Subjective Markers of Muscle Soreness following Eccentric Exercise in Resistance-Trained Men"

_muscles, doi:10.3390/muscles2020017_

Round 1
Reviewer 1 Report
No information is provided on ethical approval. Please provide that information.
I suggest to clarify in the abstract the time after trial 1 and 3, for trial 2 and 4.
The last sentence of the abstract is a summary statement of the findings. I suggest to have a conclusion, e.g. whether there is application with respect to the findings.
Table 1 indicates skinfold measurements were used to calculate body fat(%) but BIA (L133) was used. Please revise.
Table 3. I suggest to express repetitions without decimals.
L254. Change “serval” to “several”.
L272. Change “anerobic” to “anaerobic”.
Author Response
We thank the reviewers for their time and knowledge in reviewing our manuscript. As reviewers ourselves, we understand the time it takes in truly vetting a manuscript and helping it become the best version it can be. We have attempted to appease the reviewers in every case possible and provided justifications for any comments that we thought were disputable. We hope we have improved our manuscript to the point it is suitable for publication in your journal.
Reviewer Comments
Reviewer 1
No information is provided on ethical approval. Please provide that information.
This information has been provided.
I suggest to clarify in the abstract the time after trial 1 and 3, for trial 2 and 4.
This has been clarified as suggested
The last sentence of the abstract is a summary statement of the findings. I suggest to have a conclusion, e.g. whether there is application with respect to the findings.
Thank you for this suggestion. However, provided we are currently 50+ words beyond the current abstract limit, we kept the conclusion statements to just the discussion of the paper.
Table 1 indicates skinfold measurements were used to calculate body fat (%) but BIA (L133) was used. Please revise.
This has been changed as suggested. Thank you for pointing this out.
Table 3. I suggest to express repetitions without decimals.
Changed as suggested.
L254. Change “serval” to “several”.
Changed as suggested.
L272. Change “anerobic” to “anaerobic”.
Changed as suggested.
Reviewer 2 Report
The article "Astaxanthin Supplementation Reduces Subjective Markers of Muscle Soreness Following Eccentric Exercise in Resistance-Trained Men" is of huge interest to readers. And the authors did great work with it. However, I found several issues that the authors may make better for readers.
1: L 45-46, the "however" after "...DOMS (4)" seems not to be necessary. Please delete it.
2: L 83, the authors should put "." after "...a needle (27)"
3: This reviewer strongly recommends the authors make a figure of the study protocol for L 128-131. The results are a bit confusing.
4: Table 1, This reviewer recommends the authors remove the "Skinfold" because the word "Skinfold" is frequently used for data with a caliper.
5: L 162-163, "Subjects were then stratified into groups based on the 48 hours post VAS of trial by an independent investigator not involved in the data collection." How did the researcher divide subjects into two groups based on the VAS? Please clarify it more.
6: L165, Please consistently abbreviate the "performance testing" as "PT." In L 217, the "PT" suddenly shows up.
7: Tables 2 & 4, Please consistently abbreviate "Astaxanthin" as "AX" and "Placebo" as "PLA." And please add a footnote to Table 3 same as Tables 2 & 4.
8: All tables, Please clarify the sample size of each group.
9: L 259-267, This reviewer could not get why the authors emphasize the discrepancies in EIMD protocol, including eta squared. The eta squared has been reported to be inappropriate to represent effect size (PMID: 14664681). This reviewer thinks that the authors should discuss more rather than EIMD why while the AX supplementation reduced subjective DOMS, it did not impact PT, SPRE, or PRS.
10: This study has one major limitation. That is, there are no objective results to explain why AX supplementation reduced DOMS. Hence, most of the discussion section consists of speculation that is not based on the results of this study. Noting this limitation in the discussion section should be very important to readers. This reviewer also recommends the authors mention this limitation in the abstract for readers and future studies.
11: Moreover, there are several other limitations, such as no dietary data, especially the AX intake, no habitual training log, or a 2-arm parallel design study not a crossover. To clarify those, creating a limitation section in the discussion section would be very kind to readers.
12: There are several punctuation errors in this manuscript. Please carefully re-confirm.
Author Response
We thank the reviewers for their time and knowledge in reviewing our manuscript. As reviewers ourselves, we understand the time it takes in truly vetting a manuscript and helping it become the best version it can be. We have attempted to appease the reviewers in every case possible and provided justifications for any comments that we thought were disputable. We hope we have improved our manuscript to the point it is suitable for publication in your journal.
Reviewer Comments
Reviewer 2
The article "Astaxanthin Supplementation Reduces Subjective Markers of Muscle Soreness Following Eccentric Exercise in Resistance-Trained Men" is of huge interest to readers. And the authors did great work with it. However, I found several issues that the authors may make better for readers.
1: L 45-46, the "however" after "...DOMS (4)" seems not to be necessary. Please delete it.
Revised as suggested.
2: L 83, the authors should put "." after "...a needle (27)"
Revised as suggested.
3: This reviewer strongly recommends the authors make a figure of the study protocol for L 128-131. The results are a bit confusing.
A schematic overview figure has now been created and added as suggested.
4: Table 1, This reviewer recommends the authors remove the "Skinfold" because the word "Skinfold" is frequently used for data with a caliper.
Agreed and a mistake on our part. This has been revised and clarified.
5: L 162-163, "Subjects were then stratified into groups based on the 48 hours post VAS of trial by an independent investigator not involved in the data collection." How did the researcher divide subjects into two groups based on the VAS? Please clarify it more.
This has been detailed further.
6: L165, Please consistently abbreviate the "performance testing" as "PT." In L 217, the "PT" suddenly shows up.
We have removed this abbreviation in the manuscript for consistency purposes. It now will only be found in tables/figures.
7: Tables 2 & 4, Please consistently abbreviate "Astaxanthin" as "AX" and "Placebo" as "PLA." And please add a footnote to Table 3 same as Tables 2 & 4.
Changed as suggested.
8: All tables, Please clarify the sample size of each group.
Added as suggested.
9: L 259-267, This reviewer could not get why the authors emphasize the discrepancies in EIMD protocol, including eta squared. The eta squared has been reported to be inappropriate to represent effect size (PMID: 14664681). This reviewer thinks that the authors should discuss more rather than EIMD why while the AX supplementation reduced subjective DOMS, it did not impact PT, SPRE, or PRS.
This is a very thoughtful point and one we hope we address fully. Regarding the article provided, thank you. We were unaware of the various eta squares, specifically the general eta square as outlined in the provided PMID. First, as noted in that paper and this more recent one (PMID: 16405133), it does not state that eta square is inappropriate to represent effect size (ES), but rather that the design of a repeated measures study should dictate what type of ES should be chosen. In general, it appears that the general eta square is best as it does not overestimate ES like partial eta square and allows for ES comparisons across many different study types. This is great to know moving forward and something the lead author will incorporate for future studies. However, if the purpose of general eta square is to allow for comparisons across various studies, this is exactly what our partial eta square is allowing for in the present study. The earlier Waldman study referenced found significant differences in their subjective measures of DOMS, however, when you examine the ES in that study, it is trivial. A great example of finding a significant difference that is not meaningful for this type of study. Our team chose to report partial eta square to compare ES between similar measures found in our study and the Waldman et al study (accomplishing exactly what the authors in your referenced study suggested for the adoption of general eta squares). While the partial eta square may have overestimated ES in the Waldman study and our study, it is accomplishing the greater objective (as outlined in PMID: 16405133) which is allowing for ES comparison among like variables between our study and theirs. This is our justification for why we adopted a leg-press EIMD protocol compared to the drop jump protocol found in the Waldman study. Although their drop jumps induced “significant” soreness, the magnitude was quite small. We therefore chose to adopt a protocol that would not only hopefully induce significant subjective soreness, but the magnitude would be much greater which is what was accomplished in the present study. Finally, we have also elaborated further in the discussion for our speculation (since we had no objective measures as outlined later) for why there were no changes in our remaining variables.
10: This study has one major limitation. That is, there are no objective results to explain why AX supplementation reduced DOMS. Hence, most of the discussion section consists of speculation that is not based on the results of this study. Noting this limitation in the discussion section should be very important to readers. This reviewer also recommends the authors mention this limitation in the abstract for readers and future studies.
Agreed, and this has been noted in the discussion and is a part of our limitations section.
11: Moreover, there are several other limitations, such as no dietary data, especially the AX intake, no habitual training log, or a 2-arm parallel design study not a crossover. To clarify those, creating a limitation section in the discussion section would be very kind to readers.
We have created a limitations section and addressed these issues except the 2-arm design. The study was modeled off previous work that used between subjects parallel designs (PMID: 31025894, 27051592) and is line with recent supplement testing guidelines (PMID: 29468949). The 2-arm design is usually weakened by a potential order effect, but we attempted to minimize this effect by performing a group stratification based on 48h post VAS from trial one to balance the groups based on initial muscle damage soreness.
12: There are several punctuation errors in this manuscript. Please carefully re-confirm.
The research team and two individuals not a part of the study have read over the current paper and checked/corrected for grammatical/spelling errors.
Reviewer 3 Report
The present investigation has the following major methodological limitations.
In the study was used a 2-arm parallel study design, which could be not suitable for nutritional interventions since the response to eccentric exercise could be different between subjects. So, the effect of the nutritional supplementation could affect muscle damage indices randomly (i.e., either positive or negative).
The same subjects performed the damaging exercise trials pre and post nutritional supplementation. However, this design suffers the repeated bout effect phenomenon (PMID: 18172668; 32170443). This means that the exercise post supplementation would cause lower alterations in muscle damage indices due to adaptations derived from the first damaging exercise.
Author Response
We thank the reviewers for their time and knowledge in reviewing our manuscript. As reviewers ourselves, we understand the time it takes in truly vetting a manuscript and helping it become the best version it can be. We have attempted to appease the reviewers in every case possible and provided justifications for any comments that we thought were disputable. We hope we have improved our manuscript to the point it is suitable for publication in your journal.
Reviewer Comments
Reviewer 3
The present investigation has the following major methodological limitations.
In the study was used a 2-arm parallel study design, which could be not suitable for nutritional interventions since the response to eccentric exercise could be different between subjects. So, the effect of the nutritional supplementation could affect muscle damage indices randomly (i.e., either positive or negative). The same subjects performed the damaging exercise trials pre and post nutritional supplementation. However, this design suffers the repeated bout effect phenomenon (PMID: 18172668; 32170443). This means that the exercise post supplementation would cause lower alterations in muscle damage indices due to adaptations derived from the first damaging exercise.
Thank you for the comment. We agree that this is a limitation and have noted at this in our updated limitations section, however, the study was modeled off previous work that used between subjects parallel designs (PMID: 31025894, 27051592) and is line with recent supplement testing guidelines (PMID: 29468949). We attempted to minimize this effect by performing a group stratification based on 48h post VAS from trial one to balance the groups based on initial muscle damage soreness. Moreover, it is important to note that the results of our study suggest that the placebo group actually did not experience a reduction in post supplementation VAS or SORE which suggests that the repeated bout effect was not present in the placebo group. In addition, we would argue that the alternative to our approach (within design with crossover), would have exacerbated the potential repeated bout effect even further provided our subjects would have been exposed to the trials twice as much as they were in the current design. While the authors agree with the potential limitation with respect to muscle soreness, it is important to note that a secondary aim of the study was to examine the impact of supplementation on resistance exercise performance; thus, it was necessary to include pre supplementation assessments of these procedures.
Reviewer 4 Report
The study is dedicated to investigation of the astaxanthin effect on delayed-onset muscle soreness (DOMS). The authors have chosen the visual analogue scale (VAS) as the only measure of DOMS, and to my point of view this is the critical weakeness of the study. The authors cite the research by Kanda and co-authors, where it was shown that VAS correlates with myoglobin concentration (r=0.73). This statement cannot be the base to make an assumption that VAS could be the only measure of DOMS or that VAS changes correlate with ROS changes, so the most of the discussion section tends to be unconfirmed by the results of this study.
Author Response
Reviewer Comments
- The study is dedicated to investigation of the astaxanthin effect on delayed-onset muscle soreness (DOMS). The authors have chosen the visual analogue scale (VAS) as the only measure of DOMS, and to my point of view this is the critical weakeness of the study. The authors cite the research by Kanda and co-authors, where it was shown that VAS correlates with myoglobin concentration (r=0.73). This statement cannot be the base to make an assumption that VAS could be the only measure of DOMS or that VAS changes correlate with ROS changes, so the most of the discussion section tends to be unconfirmed by the results of this study.
Thank you for your comments. We chose to use a visual analogue scale (VAS) and a delayed-inset muscle soreness (DOMS) questionnaire labeled as SORE as our primary measures due to their use in previous studies with either astaxanthin (AX) or similar antioxidants (1-3). In addition, previous work from our lab did use blood biomarkers with AX supplementation, but it was intentional that we used subjective markers in this present study, as the practicality would be greater for those coaches/athletes alike who generally don’t have access to blood biomarkers or some of the more sophisticated techniques for assessing DOMS. We do agree that VAS and SORE are not the only measures of DOMS and have listed the lack of objective measures in our limitation sections. Additionally, we also agree that that changes in VAS and SORE do not correlate with changes in ROS and we have corrected this paper to read as such. This has been updated in the manuscript, particularly in the discussion section to avoid this assumption. While VAS and SORE do not correlate with a reduction in ROS, these measures do signify a reduction in perceived soreness caused by DOMS. This point has been clarified more clearly in the updated manuscript.
Refs:
- Waldman, H. S., Bryant, A. R., Parten, A. L., Grozier, C. D., & McAllister, M. J. (2022). Astaxanthin Supplementation Does Not Affect Markers of Muscle Damage or Inflammation After an Exercise-Induced Muscle Damage Protocol in Resistance-Trained Males. The Journal of Strength & Conditioning Research, 10-1519.
- Ms, S. A. B., Waldman, PhD, H. S., Krings, PhD, B. M., Lamberth, PhD, J., Smith, PhD, J. W., & McAllister, PhD, M. J. (2020). Effect of curcumin supplementation on exercise-induced oxidative stress, inflammation, muscle damage, and muscle soreness. Journal of Dietary Supplements, 17(4), 401-414.

Round 2
Reviewer 2 Report
Thanks for your modification and correction.
This reviewer is sure that the quality of this MS has been improved.
There is one minor issue as below. Please consider.
1. Figure 1: A line overlaps the square box (Trial 3). And please re-confirm the location of the arrows. Those are random locations.
Author Response
Reviewer Comments
Reviewer 2
Thanks for your modification and correction. This reviewer is sure that the quality of this MS has been improved. There is one minor issue as below. Please consider.
- Figure 1: A line overlaps the square box (Trial 3). And please re-confirm the location of the arrows. Those are random locations.
Revised as suggested. Thank you for pointing this out. We think this may be a formatting issue with the journal and if accepted, will provide the journal with the direct figure rather than placing it directly into the manuscript.

Reviewer 3 Report
The repeated bout effect is a major methodological problem. The fact that a 2-arm study design was used, the differences that were observed between the two groups could be attributed to the supplementation but also to a number of reasons except the supplementation.
Author Response
Reviewer Comments
Reviewer 3
- The repeated bout effect is a major methodological problem. The fact that a 2-arm study design was used, the differences that were observed between the two groups could be attributed to the supplementation but also to a number of reasons except the supplementation.
Thank you for the comments. We do acknowledge that the repeated bout effect could be a limitation and thus, we have added this to the limitations section. However, we would like to point out several observations. First, the repeated bout affect has been primarily seen in untrained populations (1,2). Second, the data is fairly mixed with regard to the repeated bout effect in trained males. There is quite a bit of data demonstrating that the repeated bout effect is minimal or non-existent in trained males (3-5) and other data demonstrating that if the repeated bout effect does occur in trained males, it’s within a 4-week period (6). Not only did we use trained males, but our post testing occurred after 4-weeks of supplementation. Both are factors that should have minimized a presence of a repeated bout effect. In addition, this study’s methodology is arguably a strong design provided were able to stratify these groups based on DOMS initially, rather than true randomization to a treatment group with only post-testing. This could have easily led to one group reporting initially higher levels of DOMS compared to another group and would have provided a bigger limitation in our opinion. However, by stratifying both groups initially and ensuring there were no differences in DOMS going into the supplementation period, we ensured that changes that took place during the post-testing period was in fact due to the supplement rather than a repeated bout effect. This, coupled with the placebo group not experiencing a reduction in VAS or SORE post supplementation and the group stratification based on 48h post VAS to balance the groups based on initial muscle damage, leads us to believe that the differences in soreness are attributed to the supplementation.
Refs:
- Barroso, R., Roschel, H., Ugrinowitsch, C., Araujo, R., Nosaka, K., & Tricoli, V. (2010). Effect of eccentric contraction velocity on muscle damage in repeated bouts of elbow flexor exercise. Applied Physiology, Nutrition, and Metabolism, 35(4), 534-540.
- Chen, T. C., Chen, H. L., Lin, M. J., Wu, C. J., & Nosaka, K. (2009). Muscle damage responses of the elbow flexors to four maximal eccentric exercise bouts performed every 4 weeks. European journal of applied physiology, 106, 267-275.
- Falvo, M. J., Schilling, B. K., Bloomer, R. J., Smith, W. A., & Creasy, A. C. (2007). Efficacy of prior eccentric exercise in attenuating impaired exercise performance after muscle injury in resistance trained men. The Journal of Strength & Conditioning Research, 21(4), 1053-1060.
- Falvo, M. J., Schilling, B. K., Bloomer, R. J., Smith, W. A., & Creasy, A. C. (2007). Efficacy of prior eccentric exercise in attenuating impaired exercise performance after muscle injury in resistance trained men. The Journal of Strength & Conditioning Research, 21(4), 1053-1060.
- Falvo, M. J., Schilling, B. K., Bloomer, R. J., & Smith, W. A. (2009). Repeated bout effect is absent in resistance trained men: an electromyographic analysis. Journal of Electromyography and Kinesiology, 19(6), e529-e535.
- Meneghel, A. J., Verlengia, R., Crisp, A. H., Aoki, M. S., Nosaka, K., Da Mota, G. R., & Lopes, C. R. (2014). Muscle damage of resistance-trained men after two bouts of eccentric bench press exercise. The Journal of Strength & Conditioning Research, 28(10), 2961-2966.

Reviewer 4 Report
The discussion section is improved, however, the conclusion of the study "our findings suggest that 4-weeks of AX supplementation <....> significantly reduced DOMS following eccentric exercise" should be corrected, as only subjective perception of DOMS was measured.
Author Response
Reviewer 4
The discussion section is improved, however, the conclusion of the study "our findings suggest that 4-weeks of AX supplementation <....> significantly reduced DOMS following eccentric exercise" should be corrected, as only subjective perception of DOMS was measured.
This is an accurate point and one we have corrected in our conclusion statement. Thank you for your time and reviewing our paper. Your comments have improved our paper thoroughly.
Round 3
Reviewer 3 Report
The present investigation has crucial methodological limitations. Specifically, the use of 2-arm study design that was used and the repeated bout effect phenomenon which may has affected the results of the present investigation.
Author Response
Reviewer 3
The present investigation has crucial methodological limitations. Specifically, the use of 2-arm study design that was used and the repeated bout effect phenomenon which may has affected the results of the present investigation.
Thank you for your comments. Provided this is your 3rd comment suggesting the same limitation, we must agree to disagree. There are no further changes this study can make to satisfy your comment. In our last round of revisions, we provided you extensive justification for our study design, including the fact that our study controlled for the potential of a repeated bout effect with our stratification. Overall, we do appreciate your time in reviewing our manuscript and dedication to maintain rigorous and valid research.